# Oral Iron Absorption of Ferric Citrate Hydrate and Hepcidin-25 in Hemodialysis Patients: A Prospective, Multicenter, Observational Riona-Oral Iron Absorption Trial

**DOI:** 10.3390/ijms241813779

**Published:** 2023-09-07

**Authors:** Naohisa Tomosugi, Yoshitaka Koshino, Chie Ogawa, Kunimi Maeda, Noriaki Shimada, Kimio Tomita, Shoichiro Daimon, Tsutomu Shikano, Kazuyuki Ryu, Toru Takatani, Kazuya Sakamoto, Satonori Ueyama, Daisuke Nagasaku, Masato Nakamura, Shibun Ra, Masataka Nishimura, Chieko Takagi, Yoji Ishii, Noritoshi Kudo, Shinsuke Takechi, Takashi Ishizu, Takamoto Yanagawa, Masamichi Fukuda, Yutaka Nitta, Takayuki Yamaoka, Taku Saito, Suzuko Imayoshi, Momoyo Omata, Joji Oshima, Akira Onozaki, Hiroaki Ichihashi, Yasuhisa Matsushima, Hisahito Takae, Ryoichi Nakazawa, Koichi Ikeda, Masato Tsuboi, Keiko Konishi, Shouzaburo Kato, Maki Ooura, Masaki Koyama, Tsukasa Naganuma, Makoto Ogi, Shigeyuki Katayama, Toshiaki Okumura, Shigemi Kameda, Sayuri Shirai

**Affiliations:** 1Division of Systems Bioscience for Drug Discovery, Project Research Center, Medical Research Institute, Kanazawa Medical University, Kahoku 920-0293, Ishikawa, Japan; 2Mizuho Hospital, Tsubata 929-0346, Ishikawa, Japan; mizuho@tiara.ocn.ne.jp; 3Maeda Institute of Renal Research Musashikosugi, Kawasaki 211-0063, Kanagawa, Japan; tato.ogawa@gmail.com; 4Maeda Institute of Renal Research Shakujii, Nerima 177-0041, Tokyo, Japan; kuni@maeda-irr.com; 5Tachibana Clinic, Sumida 131-0043, Tokyo, Japan; noriaki.shimada@tachibana-cl.or.jp; 6The Chronic Kidney Disease Research Center, Tomei Atsugi General Hospital, Atsugi 243-8571, Kanagawa, Japan; tomyw23@nifty.com; 7Department of Nephrology, Daimon Clinic for Internal Medicine, Nonoichi 921-8802, Ishikawa, Japan; dai-clinic@m2.spacelan.ne.jp; 8Kyoto Okamoto Memorial Hospital, Kuze 613-0034, Kyoto, Japan; sktu@okamoto-hp.or.jp (T.S.); ryu.kazuyuki@gmail.com (K.R.); 9Nephrology Division, Tojinkai Hospital, Fushimi 612-8026, Kyoto, Japan; takatani@tojinkai.jp; 10Department of Urology, Tomakomai Nisshou Hospital, Tomakomai 053-0803, Hokkaido, Japan; k.sakamoto@nisshou-hospital.jp; 11Jinaikai Ueyama Hospital, Kagoshima 890-0073, Kagoshima, Japan; s-ueyama@jin-ai-kai.or.jp; 12Yujin-Yamazaki Hospital, Hikone 522-0044, Shiga, Japan; nagasaku@simosaka.jp; 13Susono Daiichi Clinic, Susono 410-1112, Shizuoka, Japan; susonoce@joy.ocn.ne.jp; 14Noheji Clinic, Noheji 039-3152, Aomori, Japan; kashiranodagashira@yahoo.co.jp; 15Shimosaka Clinic, Nagahama 526-0044, Shiga, Japan; nishimura@simosaka.jp; 16Ohgo Clinic, Maebashi 371-0232, Gunma, Japan; chiekot@cj8.so-net.ne.jp; 17Nozatomon Clinic, Himeji 670-0011, Hyogo, Japan; touseki@nozatomon.jp; 18Kowa Clinic, Goshogawara 037-0066, Aomori, Japan; kowa@hakuseikai.com; 19Takechi Clinic, Iyo 791-3141, Ehime, Japan; ikei@takechi-clinic.jp; 20Department of Nephrology, Tsukuba Central Hospital, Ushiku 300-1211, Ibaraki, Japan; takasi.isizu@gmail.com (T.I.); yana_3173@yahoo.co.jp (T.Y.); 21Iwakuni Medical Center, Iwakuni 740-0021, Yamaguchi, Japan; mami9380@yahoo.co.jp; 22The Department of Nephrology, Saiseikai Shimonoseki General Hospital, Shimonoseki 759-6603, Yamaguchi, Japan; y-nitta@simo.saiseikai.or.jp (Y.N.); simosai.jinnai.2018@gmail.com (T.Y.); 23Saito Memorial Hospital, Kawaguchi 332-0034, Saitama, Japan; saitoukinen@med.email.ne.jp (T.S.); papamia888@yahoo.co.jp (S.I.); 24Department of Internal Medicine, Hachioji Azumacho Clinic, Hachioji-shi 192-0082, Tokyo, Japan; m-omata@juno.ocn.ne.jp; 25Kubojima Clinic, Kumagaya 360-0831, Saitama, Japan; jo123kum@sakitama.or.jp; 26Tokatsu-Clinic Hospital, Matsudo 271-0067, Chiba, Japan; anzen.tch@mbr.nifty.com; 27Tokatsu Clinic Yabashira, Matsudo 270-2253, Chiba, Japan; tc-yahashira@nifty.com; 28Tokatsu Clinic Kashiwa, Kashiwa 277-0005, Chiba, Japan; tc-kasiwa@nifty.com; 29Tokatsu Clinic Matsudo, Matsudo 271-0077, Chiba, Japan; tc-matudo@nifty.com; 30Tokatsu Clinic Mirai, Matsudo 271-0091, Chiba, Japan; r.nakazawa@mbr.nifty.com; 31Tokatsu Clinic Koiwa, Edogawa 133-0056, Tokyo, Japan; tomosuginaohisa@gmail.com; 32Kaikoukai Anjo Kyoritsu Clinic, Anjo 446-0065, Aichi, Japan; m-tsuboi@kaikou.or.jp; 33Seiwa Hospital, Toyama 931-8431, Toyama, Japan; seiwakeiko7002@yahoo.co.jp; 34Nishi Interchange Clinic for Internal Medicine and Dialysis, Kanazawa 921-8001, Ishikawa, Japan; info@west-inter.or.jp; 35Maro Clinic, Tanabe 646-0004, Wakayama, Japan; maroclinic339@gmail.com; 36Nishijin Hospital, Kyoto 602-8319, Kyoto, Japan; koyama580@nisijin.net; 37Department of Nephrology, Yamanashi Prefectural Central Hospital, Kofu 400-0027, Yamanashi, Japan; naganuma-bfpn@ych.pref.yamanashi.jp; 38Department of Internal Medicine, Yuurinkouseikai Fuji Hospital, Gotemba 412-0043, Shizuoka, Japan; ogi@yuurinkouseikai.or.jp; 39Katayama Clinic, Iwakuni 741-0072, Yamaguchi, Japan; ktclinic@bronze.ocn.ne.jp; 40Mizue Yuai Clinic, Edogawa 133-0065, Tokyo, Japan; yuai@zuiko.or.jp; 41Joetsu General Hospital, Joetsu 943-8507, Niigata, Japan; kamedash@joetsu-hp.jp; 42Division of Nephrology and Hypertension, Department of Internal Medicine, St. Marianna University Yokohama Seibu Hospital, Yokohama 241-0811, Kanagawa, Japan; sirababu@marianna-u.ac.jp

**Keywords:** oral iron absorption, ferric citrate hydrate, hemodialysis, hepcidin-25, iron shift

## Abstract

Oral ferric citrate hydrate (FCH) is effective for iron deficiencies in hemodialysis patients; however, how iron balance in the body affects iron absorption in the intestinal tract remains unclear. This prospective observational study (Riona-Oral Iron Absorption Trial, R-OIAT, UMIN 000031406) was conducted at 42 hemodialysis centers in Japan, wherein 268 hemodialysis patients without inflammation were enrolled and treated with a fixed amount of FCH for 6 months. We assessed the predictive value of hepcidin-25 for iron absorption and iron shift between ferritin (FTN) and red blood cells (RBCs) following FCH therapy. Serum iron changes at 2 h (ΔFe2h) after FCH ingestion were evaluated as iron absorption. The primary outcome was the quantitative delineation of iron variables with respect to ΔFe2h, and the secondary outcome was the description of the predictors of the body’s iron balance. Generalized estimating equations (GEEs) were used to identify the determinants of iron absorption during each phase of FCH treatment. ΔFe2h increased when hepcidin-25 and TSAT decreased (−0.459, −0.643 to −0.276, *p* = 0.000; −0.648, −1.099 to −0.197, *p* = 0.005, respectively) in GEEs. FTN increased when RBCs decreased (−1.392, −1.749 to −1.035, *p* = 0.000) and hepcidin-25 increased (0.297, 0.239 to 0.355, *p* = 0.000). Limiting erythropoiesis to maintain hemoglobin levels induces RBC reduction in hemodialysis patients, resulting in increased hepcidin-25 and FTN levels. Hepcidin-25 production may prompt an iron shift from RBC iron to FTN iron, inhibiting iron absorption even with continued FCH intake.

## 1. Introduction

Patients undergoing hemodialysis lose 1.5–3.0 g of iron annually due to dialysis and periodic laboratory evaluations [1], leading to iron deficiency. The oral supplementation of ferrous iron is inconvenient for iron deficiency anemia in dialysis patients due to gastrointestinal side effects [2], while the use of highly soluble ferric citrate (FC) is increasing because it has fewer side effects [3]. The long-term administration of FC increases ferritin (FTN) and transferrin saturation (TSAT), reduces intravenous iron and erythropoiesis-stimulating agent (ESA) dose requirements, and maintains hemoglobin (Hb) levels in hemodialysis patients [4,5,6]. Nevertheless, it remains unclear what fraction of FC is absorbed, what type of iron state promotes iron absorption, and whether long-term FC administration causes iron overload.

FTN is an index of stored iron [7], but its appropriate value in the body remains unknown. There are large differences in FTN levels in hemodialysis patients across countries, with patients in Japan and the USA having the lowest and highest FTN levels, respectively [8]. It is speculated that if a large difference exists in the amount of iron stored, there will be a difference in iron absorption regulated by hepcidin-25; however, previous FC studies have reported the same phenomena [4,5]. Yokoyama et al. [4] found that FTN essentially plateaued at week 28, increasing from 57 ng/mL at baseline to 227 ng/mL after 52 weeks, whereas ESA doses gradually decreased during this period. Moreover, Lewis et al. [5] reported that FTN levels increased from 593 ng/mL at baseline to 899 ng/mL at 52 weeks, plateauing at 6 months, while the ESA dose decreased in FC studies.

FTN correlates strongly and positively with hepcidin-25, an iron regulatory hormone, as measured using surface-enhanced laser desorption/ionization–time-of-flight mass spectrometry, liquid chromatography–tandem mass spectroscopy (LC-MS/MS), and enzyme-linked immunosorbent assay [9,10,11]. Thus, iron states with high FTN levels were expected to be associated with high hepcidin expression. At higher serum hepcidin-25 levels, hepcidin-25 binds to the iron exporter ferroportin (FPN) on the basolateral surface of duodenal enterocytes, thereby inducing its internalization and degradation and blocking intestinal iron absorption [12]. However, the details of this process are unclear because iron absorption and serum hepcidin-25 levels were not determined in these studies [4,5,6].

Moreover, whether long-term FC administration induces an iron overload remains unknown. The plateauing of FTN levels at 6 months despite continued FC administration may be related to the strict regulation and saturation of iron absorption [6]. However, in addition to the hepcidin–ferroportin axis and the erythroferrone produced during erythroblast production [13] or the guideline-based arbitrary reduction in the ESA dose that regulates red blood cell (RBC) production, no iron-regulatory metabolic mechanism is currently known. Iron metabolism is closely linked to erythropoiesis [14], and inadequate erythropoiesis may disturb iron metabolism. However, the precise mechanisms that regulate iron flow in patients undergoing hemodialysis with ESA therapy are unknown. Hence, we conducted the Riona-Oral Iron Absorption Trial (R-OIAT) using ferric citrate hydrate (FCH; Riona, Torii Pharmaceutical Co. Ltd., Tokyo, Japan) as an analytical, prospective, observational study to intercept the crosstalk between erythropoiesis and iron metabolism using intestinal iron absorption and serum hepcidin-25 as the main indices. In this study, we did not aim to supplement iron but rather to observe iron states in patients who received a fixed amount of FCH as a phosphate binder for 6 months.

## 2. Results

### 2.1. Baseline Characteristics of the Study Participants

A total of 268 patients were included in the R-OIAT (Figure 1). Between 1 June 2017 and 1 June 2019, 584 hemodialysis patients using FCH as a phosphate binder were enrolled in the R-OIAT. According to the exclusion criteria, 213 and 78 participants were excluded at 3 months (M3) and 6 months (M6), respectively. The numbers of patients with a serum C-reactive protein (CRP) level > 0.5 mg/dL at baseline (M0), M3, and M6 were 103, 37, and 25, respectively. Finally, 268 patients who continued to receive a fixed amount of FCH for 6 months were analyzed. The trial concluded in June 2020. The baseline characteristics of the participants are shown in Table 1. The mean age of the participants was 63.0 years (SD, 11.7), 57% of patients were male, 55.6% received 750 mg of FCH, and 49.6% were injected with darbepoetin alfa. 

### 2.2. Association of Changes in Clinical Indices at Different 3-Month Intervals

The clinical measurements on the day of R-OIAT testing at M0, M3, and M6 are summarized in Table 2. FTN levels increased significantly from 100.7 ng/mL at baseline to 116.7 ng/mL at 24 weeks, while the ESA dose decreased, and Hb concentrations remained stable. The effects of the amount of FCH on the changes in iron absorption (ΔFe2h) and iron variables for 6 months are shown in Table 3. In patients receiving 750 mg (*n* = 149), 1500 mg (*n* = 101), or 2250 mg (*n* = 18) of FCH, no significant interaction between the amount of FCH and the exposure time was observed for all iron variables; this means that three independent groups changed similarly to each other with respect to time (Table 3). Although ΔFe2h levels were significantly lower in the low-dose group than in the high-dose group, the differences remained mostly unchanged throughout the 6 months. FTN and hepcidin-25 levels were significantly lower in the low-dose group than in the high-dose group at M0, and the difference was maintained at M3 and M6. In each group, there was a significant increase in FTN, Hb, MCH, and TSAT levels over 6 months. When RBCs ≤ 350 × 10^4^ /μL, the number of cases with Hb exceeding 12 g/dL was only two (0.7%).

### 2.3. Association of ΔFe2h with Iron Variables

Generalized estimating equations (GEEs) were used to identify the determinants of ΔFe2h during each phase of FCH treatment: M0, M3, and M6. ΔFe2h was affected by hepcidin-25 and TSAT (Table 4). Log ΔFe2h increased by 0.459 μg/dL per 1 ng/mL decrease in Log hepcidin-25 [95% CI: −0.643 to −0.276, *p* = 0.000] and increased by 0.648 μg/dL per 1 % decrease in Log TSAT [95% CI: −1.099 to −0.197, *p* = 0.005]. Hb levels were excluded because of collinearity with respect to RBCs based on the formula Hb = RBC × MCH. In the three datasets of iron markers per patient at M0, M3, and M6 (*n* = 804), there were significant negative correlations of ΔFe2h with hepcidin-25 (r = −0.163, *p* = 0.000, Figure 2A) and TSAT (r = −0.149, *p* = 0.000, Figure 2B).

### 2.4. Association of Hepcidin-25 with Iron Variables

Predictors for hepcidin-25 were analyzed via GEEs using iron variables, excluding ΔFe2h (Table 5). Hepcidin-25 was affected by RBC, MCH, TSAT, FTN, and ESA. Log hepcidin-25 decreased by 0.790 ng/mL per 1 × 10^4^/μL decrease in Log RBC [95% CI: 0.178 to 1.401, *p* = 0.011]; by 1.667 ng/mL per 1 pg/cell decrease in Log MCH [95% CI: 0.121 to 3.213, *p* = 0.035]; by 0.372 ng/mL per 1 % decrease in Log TSAT [95% CI: 0.144 to 0.600, *p* = 0.001]; by 0.754 ng/ml per 1 ng/mL decrease in Log FTN [95% CI: 0.637 to 0.871, *p* = 0.000]; and by 0.353 ng/mL per 1 IU/week increase in Log ESA [95% CI: −0.462 to −0.245, *p* = 0.000]. In the three datasets of iron markers per patient at M0, M3, and M6 (*n* = 804), there were significant correlations between hepcidin-25 and ESA (r = 0.201, *p* = 0.000, Figure 3A), FTN (r = 0.646, *p* = 0.000, Figure 3B), and TSAT (r = 0.278, *p* =0.000, Figure 3D). An increase in FTN up to 100 ng/mL was accompanied by an increase in hepcidin-25, but even if FTN increased beyond that level, no increase in hepcidin-25 was observed, as if there was an upper limit for hepcidin expression. MCH had no significant correlations with hepcidin-25 (r = 0.065, *p* = 0.063, Figure 3C).

### 2.5. Patterns of FTN Fluctuations

Contrary to expectations, because FTN did not necessarily increase gradually with the continuous administration of FCH, the fluctuation pattern of FTN was analyzed. Patients were classified into four groups based on 3-month changes in FTN from M0 to M3 or from M3 to M6 (denoted as ΔFTN_M3-M0_ and ΔFTN_M6-M3_, respectively): P-1 (ΔFTN_M3-M0_: positive; ΔFTN_M6-M3_: positive; *n* = 63), P-2 (ΔFTN_M3-M0_: positive; ΔFTN_M6-M3_: negative; *n* = 84), P-3 (ΔFTN_M3-M0_: negative; ΔFTN_M6-M3_: positive; *n* = 84), and P-4 (ΔFTN_M3-M0_: negative; ΔFTN_M6-M3_: negative; *n* = 37). FTN gradually increased only in 63 cases (23.5%) during both 3-month intervals (Figure 4) and gradually decreased in 37 cases (13.8%). Interestingly, an increase in FTN corresponded to a significant decrease in RBCs and Hb and vice versa with the mirror image in each group. Moreover, there was a significant negative correlation between ΔFTN_M3-M0_ and ΔFTN_M6-M3_ (r = 0.571, Figure 5A). In total, 84 of 147 patients (57.1%) whose ΔFTN_M3-M0_ was positive were negative with respect to ΔFTN_M6-M3_, and 84 of 121 patients (69.4%) whose ΔFTN_M3-M0_ was negative were positive with respect to ΔFTN_M6-M3_. Similar negative correlations were observed between the changes in the first and second 3-month periods with respect to hepcidin-25, Hb, and RBCs (r = 0.527, 0.589, and 0.599, respectively; Figure 5B–D). 

### 2.6. Background of Changes in FTN

Predictors for FTN were also analyzed via GEEs using iron variables, excluding ΔFe2h (Table 5). FTN was affected by hepcidin-25, RBC, and TSAT. Log FTN increased by 0.297 ng/mL per 1 ng/mL increase in Log hepcidin-25 [95% CI: 0.239 to 0.355, *p* = 0.000]; by 1.392 ng/mL per 1 × 10^4^/μL decrease in Log RBCs [95% CI: −1.749 to −1.035, *p* = 0.000]; and by 0.233 ng/mL per 1 % increase in Log TSAT [95% CI: 0.099 to 0.367, *p* = 0.001]. 

To analyze the effect of the change in RBCs counts on FTN values over 3 months, the levels at M0 and M3 were treated as starting points for the first and second 3-month periods, and levels at M3 and M6 were treated as 3-month points, respectively. Two datasets of iron variables were evaluated equally, and the changes in iron variables over 3 months were represented as Δ_3M_ (*n* = 536). Each datum was classified into four groups, G-1, G-2, G-3, or G-4, according to the RBC value at start points M0 or M3; RBC ≤ 300: 300 < RBC ≤ 350, 350 < RBC ≤ 400, and RBC > 400 × 10^4^/μL, respectively. Furthermore, each case was classified into G-a when each ΔRBC_3M_ was negative and into G-b when ΔRBC_3M_ was positive. The start and end points were vectorized as the mean values of MCH and RBCs (Figure 6A). In any RBC region, changes in FTN for 3 months in each group decreased in the increased RBC group (b) and increased in the decreased RBC group (a) (Figure 6B). 

## 3. Discussion

The features of this study are that participants were taking a fixed amount of FCH without changing the dose for 6 months and had no inflammation that affected hepcidin-25 expression via IL6R [15]. Under these conditions, hepcidin-25 and TSAT were the strongest inverse explanatory factors for intestinal iron absorption based on GEEs. At the molecular level, hepcidin-25 has been shown to control the distribution density of FPN on the cell membrane’s surface [16] and intestinal divalent metal-iron transporter 1 expression [17], leading to the influx of iron from intestinal epithelial cells into the blood. More recently, Hanudel et al. [18] demonstrated that intestinal ferric citrate absorption relies on conventional enterocyte iron transport by ferroportin, without significant paracellular absorption, using Tmprss6 and FPN knockout mice. The results of this study reconfirmed that the clinical measurement of hepcidin-25 would be a great tool for estimating iron absorption in clinical practice as well. Similar phenomena may also be observed in hepatocytes and macrophages, which bear FPN on membranes [19].

MCH and TSAT were explanatory factors for hepcidin-25. MCH is calculated by dividing Hb by RBCs and indicates the average amount of Hb, including in one RBC. MCH has an upper limit of 33 [20,21], up to which erythroblasts can take up iron for Hb synthesis; thus, MCH could be taken as an indicator of iron storage capacity in RBCs. It is speculated that when the level of MCH is low—which means that there are large reserve-iron storage capacities in RBCs—serum iron may be readily supplied to erythroblasts, its concentration may decrease and iron absorption from the intestinal tract would increase to replenish it. Erythroblasts require large amounts of iron for hemoglobin synthesis; therefore, they express very high levels of TfR1 [22]. Although MCH has received less attention in the assessment of iron metabolism, it could be used to infer the reserve storage capacity of iron in the RBCs.

FTN was not an explanatory factor of iron absorption, although Eshbach et al. [23] reported that food iron absorption increases when the FTN serum is below 30 ng/mL and decreases when there is more than 100 ng/mL by evaluating total body isotope activity at 2 weeks after the ingestion of the extrinsic tag of radioiron salt added to a meal. Since the amount of iron in the blood is as low as 2–3 mg [24], it is speculated that a decrease in serum iron concentration and hepcidin-25 due to rapid iron consumption by erythropoiesis after ESA administration [25] may occur regardless of the value of stored FTN iron. Given that FPN is currently the only known iron transporter, the rapid decrease in hepcidin-25 levels after ESA administration in hemodialysis patients, which means the expansion of erythropoiesis, may be immensely advantageous for iron absorption in the intestine.

ESA and FTN were explanatory factors for hepcidin-25 in addition to TSAT, which is an index of serum iron concentrations that stimulates TfR2 in hepatocytes to induce the expression of hepcidin [26]. In our previous study [10], serum iron, hepcidin-25, and FTN levels rapidly decreased after ESA administration and then recovered during the interval for the next ESA administration. Chaston et al. [16] reported that the FPN response to hepcidin-25 differs among cells, e.g., reticuloendothelial macrophages, hepatocytes, and enterocytes, and that a rapid increase in hepcidin-25 reduces macrophage FPN expression after 2 h, but enterocyte FPN reduction appears only when hepcidin-25 levels have remained continuously high for 24 h. These phenomena may be responsible for the complicated relationship between iron absorption and hepcidin-25, and these reactions cannot be captured using one-time measurements. Time-course measurements should be included in future studies.

In the present study, the predictors of FNT were analyzed using GEEs because FTN has often been used to assess the body’s iron stores [27]. Hepcidin-25 is a positive predictor, and RBCs and ESA are negative predictors of FTN. These findings may be due to the arbitrary adjustment of ESA such that Hb could be maintained at 10–12 g/dL according to clinical guidelines [28,29]. Approximately 80% of the body’s iron is found in RBCs [30], with 90% of daily iron consumption used by erythroblasts to synthesize Hb [31]. The erythroid system maintains iron homeostasis as an iron reservoir [31] whereby iron is conserved and recycled in the body [32]. In a closed iron metabolism system, limiting erythropoiesis by reducing ESA doses could induce an increase in serum iron, followed by hepcidin-25; inhibit the delivery of iron to the blood by macrophages; and concomitantly suppress intestinal iron absorption. As a result, FTN increases in macrophages phagocytized senescent erythrocytes. This phenomenon appears to be a shift from erythrocyte iron to FTN iron or vice vers (Figure 7). Our findings provide insight into the reasons why the rate of increase in FTN is reduced at 24 weeks and why it reaches a plateau despite continuous FC exposure for 52 weeks [4,5,6]. When the Hb level is ≥12 g/dL, ESA is arbitrarily reduced, and Hb levels are adjusted between 10 and 12 g/dL according to the guidelines. As shown in Figure 8, the Hb level, which was 13 g/dL in RBCs 450 × 10^4^/μL (MCH 28.9 pg, point A), was reduced to 11.6 g/dL when reducing ESA and resetting RBCs to 400 × 10^4^/μL (point B). During this period, the FTN level increased because the RBC iron capacity decreased, and the corresponding amount of iron shifted to FTN iron. When FCH administration continued, MCH increased with iron absorption, and Hb levels easily exceeded 12 g/dL (RBCs to 400 × 10^4^/μL, MCH 32.5 pg, point C). If ESA was reduced again and RBCs reset to 350 × 10^4^/μL, the Hb level improved to 11.4 g/dL (point D) with another increase in the FTN level. When RBC is ≤350 × 10^4^/μL, there is little risk of Hb being ≥12 g/dL even if FCH therapy is continued because there is an upper limit against MCH, which falls normally within the range of 27–33 pg [20,21]. At this stage, there is no need to further reduce the ESA, and the FTN reaches a plateau without increasing. Such a series of reactions may occur regardless of the FTN value at baseline, such as 227 ng/mL in Japan and 899 ng/mL in the United States [4,5,6]. Our concern with FCH therapy is whether the long-term use of FC may cause iron overload. These findings suggest that initially setting RBC to 300–350 × 10^4^/μL followed by oral iron supplementation, in which Hb levels do not exceed 12 g/dL, would minimize fluctuations in FTN. Whether such a procedure could reduce the requirement for ESA and prevent iron overload should be clarified in future studies.

The upper limit of hepcidin-25 production in terms of the relationship with FTN has not been reported previously. We showed that the physiological concentrations of hepcidin-25 have an upper limit of more than 100 ng/mL FTN. Similar trends have been observed in myelodysplastic syndromes [33], thalassemia [34], and dialysis patients [35], but there is no description of the upper limit of hepcidin in these papers. When the cellular labile iron pool (LIP) increases, FTN expression is stimulated via iron-responsive elements and iron regulatory proteins at the translational level, and the ferrous iron of LIP is mobilized to FTN as ferric iron to reduce the risk of radical developments in the cytoplasm [7]. On the other hand, elevated LIP enhances the expression of bone morphogenetic protein 6 (BMP6) in hepatic non-parenchymal cells, which binds to the hepatocyte BMP receptor in paracrine and stimulates hepcidin expression via the BMP6/SMAD pathway to prevent iron export via FPN to blood [36]. If it is hypothesized that there is an upper limit on the levels of LIP to prevent the cytoplasm from radical exposure, BMP6 and hepcidin-25 in its downstream will also have an upper limit, and their serum levels will fluctuate according to LIP. When FTN remains undegraded in the cytoplasm, the levels of FTN might increase cumulatively without an upper limit. In myelodysplastic syndromes, patients with more than 10,000 ng/mL of FTN were reported [37]. Hepcidin-25 may be a predictor of LIP and BMP6, and cumulative increases in FTN may indicate that the LIP has repeatedly reached the upper limit. This may be the reason why FC was effective regardless of the FTN levels in previous studies [4,5,6]. Although we only targeted cases without inflammation, the difference in hepcidin synthesis via STAT3 stimulated by IL-6 during inflammation should be examined in future studies [32].

In conclusion, our data suggested that a key molecule for facilitating the crosstalk between hematopoietic and iron metabolic systems in order to keep the balance between iron consumption in serum and iron supplies to serum was hepcidin-25, and ESA was shown to be a trigger that causes a shift in iron between the body’s stored iron and stored RBC iron. Limited erythropoiesis was one of the causes of FTN increases in hemodialysis patients undergoing FCH therapy, in addition to iron absorption from the intestinal tract. A major factor altering iron-related factors in dialysis patients may be the fluctuation of RBCs, which have large capacities for iron storage. If the RBC count is maintained below 350 × 10^4^/µL, iron overload would not occur even with a sustained oral supply of iron. Although our results do not provide evidence for an adequate RBC number, our study provides a basis for clinical trials with respect to evaluating RBC-restricting strategies to minimize the ESA dose and improve iron metabolism in patients with renal anemia.

## 4. Materials and Methods

### 4.1. Study Design and Data Sources

The R-OIAT recruited patients from 42 hemodialysis centers in Japan. Eligible subjects (*n* = 584) were adult patients with end-stage renal disease who were on hemodialysis three times per week for at least one month before the R-OIAT trial and who were prescribed 750 mg–2250 mg doses of FCH as a phosphate binder. Eligible hemodialysis patients were not pregnant and had no malignancies or infections. Patients were excluded if they received intravenous iron supplementation and blood transfusion, were transferred to a different clinic, ceased using FCH, changed FCH dosage, or had inflammation (CRP level > 0.5 mg/dL). Finally, 268 individuals who received a fixed amount of FCH as a phosphate binder for 6 months were enrolled: 750 mg (*n* = 149), 1500 mg (*n* =101), and 2250 mg (*n* = 18). This study was conducted in accordance with the Declaration of Helsinki and approved by the Kanazawa Medical University Ethics Committee (No. M401). All participants provided written informed consent.

### 4.2. Procedures

The patients received FCH three times a day after meals. One tablet of FCH (250 mg) contains approximately 60 mg of ferric iron. The participants consumed their regular diets during the study period. Thus, on the day of the R-OIAT, blood was drawn early in dialysis, and patients received one-third of the total daily FCH dose shortly thereafter. After 2 h, another blood sample was obtained. The patients were on an ordinal dialysis diet for 6 months of the trial but did not take a special diet that determined the amount of protein and iron immediately prior to the iron absorption test. The patients did not fast before the test, but they did fast until the second sampling after the FCH dose. R-OIAT sampling was repeated thrice at M0, M3, and M6. Blood samples were centrifuged at 1500× *g* for 10 min, and the serum was immediately frozen at −20 °C. Samples were shipped on dry ice to Kanazawa Medical University to determine serum iron, FTN, unsaturated iron-binding capacity (UIBC), and hepcidin-25 levels. Serum iron and UIBC levels were measured using colorimetry; TSAT was calculated using the following formula: serum iron/total iron-binding capacity × 100. The FTN levels were measured using a chemiluminescent enzyme immunoassay. Serum hepcidin-25 was measured using LC-MS/MS [38] at MCProt Biotechnology (Ishikawa, Japan), with a dynamic range of 0–1000 ng/mL. LC-MS/MS was performed using 4000 QTRAP (Applied Biosystems, Foster City, CA, USA) equipped with HPLC pump (Shimadzu Corp., Kyoto, Japan). ESA doses were calculated on each oral iron absorption trial day. Because patients were taking different ESAs (i.e., epoetin (IU), darbepoetin (μg), or epoetin beta pegol (μg)), their units were converted to international units (IUs) based on previously reported data [39]. The conversion for darbepoetin and epoetin beta pegol was based on reports that found that 1 μg is equivalent to 200 IU [40]. The amount of ESAs was assessed as average weekly doses (IU/week).

### 4.3. Estimating Iron Absorption

The amount of iron absorbed after FCH administration (ΔFe2h) was determined by subtracting the serum iron concentration before administration from the serum iron concentration measured 2 h later according to previous studies [41]. In preliminary experiments, ΔFe2h ranged from 0 to 23 μg/dL in 36 control hemodialysis patients without FCH administration, and the median (inter-quartile range, IQR) of ΔFe2h was 0.5 (IQR: 0–7) μg/dL.

### 4.4. Statistical Analysis

Continuous variables were summarized as the mean and standard deviation. Categorical variables were described as the numbers and relative frequencies (%) of the subjects in each category. Iron variables were naturally log-transformed to maintain a normal distribution. Comparisons between patient groups were carried out using one-way analysis of variance (ANOVA) or repeated measures ANOVA with post hoc Bonferroni’s multiple comparison tests. A two-way repeated measures ANOVA test (condition [750 mg vs. 1500 mg vs. 2250 mg of FCH] × time [M0 vs. M3 vs. M6]) was used to analyze the effects of FCH on iron variables. Univariate correlations between variables were assessed using Pearson’s correlation test. After adjusting for independent variables, GEEs were used to identify the determinants of iron absorption during each phase of FCH treatment: M0, M3, and M6; the B coefficients with their 95% CI and the corresponding *p* values < 0.05 were used to declare the determinants of iron absorption. Independent predictors of FTN were also assessed using GEEs. Differences in iron variables between the ranges of RBCs were compared using independent *t*-tests for parametric data. All *p* values were two-tailed, with a level of <0.05. Database construction and statistical analyses were performed using Statistical Package for the Social Sciences (SPSS) version 24 (IBM SPSS Inc., Armonk, NY, USA).

## Figures and Tables

**Figure 1 ijms-24-13779-f001:**
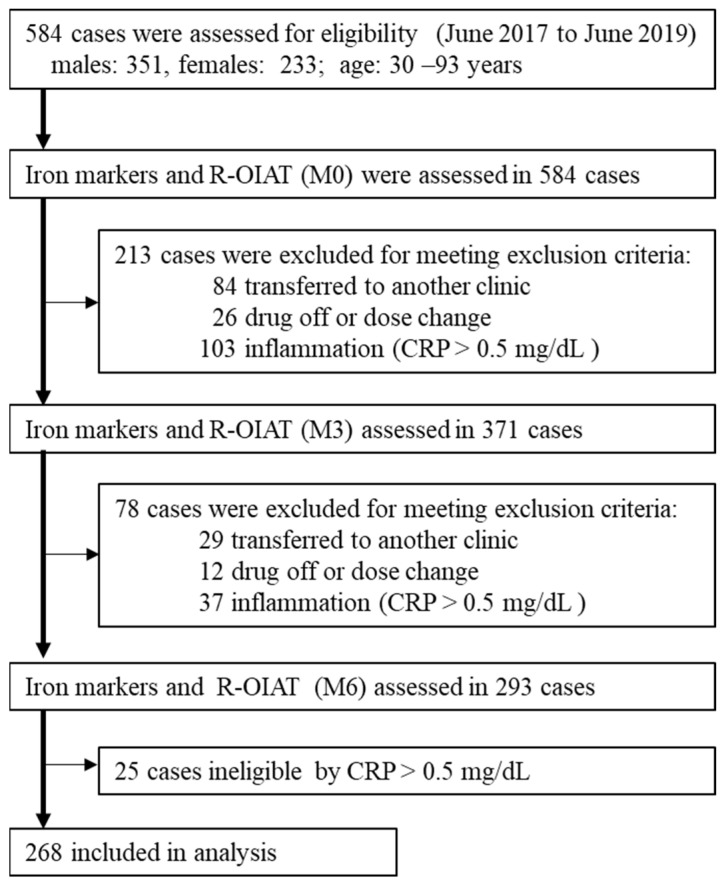
Flow chart depicting the study’s population selection.

**Figure 2 ijms-24-13779-f002:**
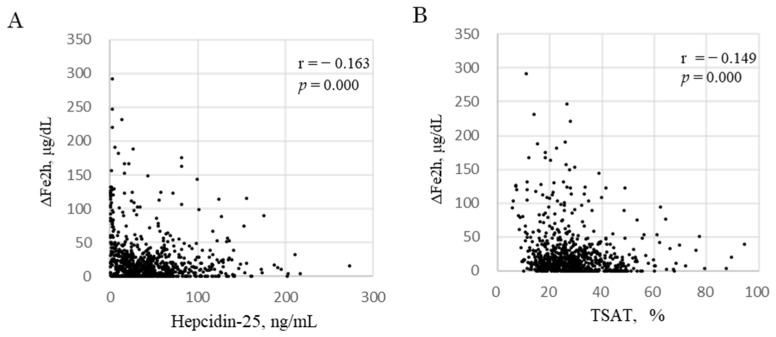
Correlation between ΔFe2h and hepcidin-25 and TSAT. Correlation between ΔFe2h and hepcidin-25 (**A**) and TSAT (**B**). The data included all samples from 268 patients at M0, M3, and M6 during FCH therapy for 6 months (black dots, *n* = 804). ΔFe2h, iron absorption; MCH, mean corpuscular hemoglobin. ΔFe2h, iron absorption in 2 h; TSAT, transferrin saturation.

**Figure 3 ijms-24-13779-f003:**
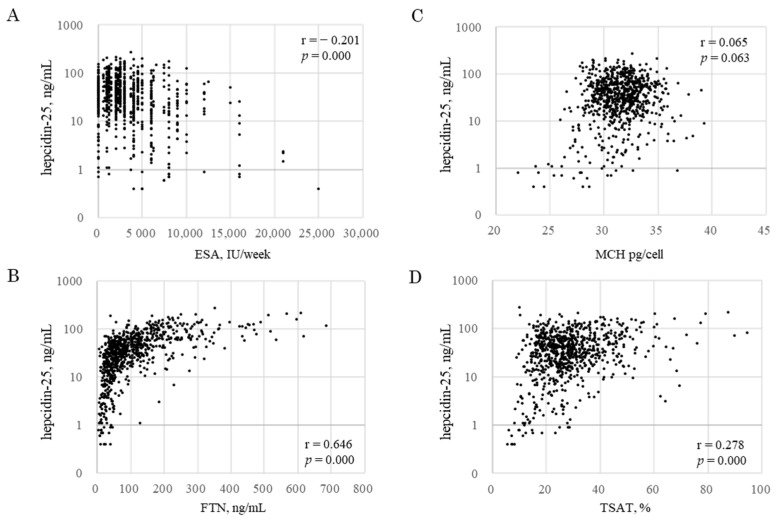
Correlation between hepcidin-25 and iron markers. Correlation between hepcidin-25 and ESA (**A**), FTN (**B**), MCH (**C**), and TSAT (**D**). Data included all samples from 268 patients at M0, M3, and M6 during FCH therapy for 6 months (block dots, *n* = 804). ESA, erythropoiesis-stimulating agent; FTN, ferritin; MCH, mean corpuscular hemoglobin; TSAT, transferrin saturation.

**Figure 4 ijms-24-13779-f004:**
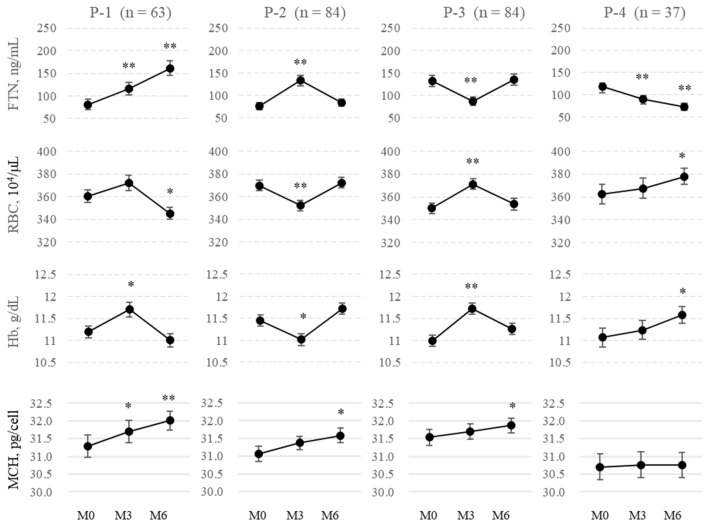
Changes in FTN and iron variables every three months (*n* = 268). Patients were classified into four groups based on 3-month changes in FTN from M0 to M3 or from M3 to M6 (denoted as ΔFTN_M3-M0_ and ΔFTN_M6-M3_, respectively); P-1 (ΔFTN_M3-M0_: positive; ΔFTN_M6-M3_: positive; *n* = 63), P-2 (ΔFTN_M3-M0_: positive; ΔFTN_M6-M3_: negative; *n* = 84), P-3 (ΔFTN_M3-M0_: negative; ΔFTN_M6-M3_: positive; *n* = 84), and P-4 (ΔFTN_M3-M0_: negative; ΔFTN_M6-M3_: negative; *n* = 37). FTN, Ferritin; RBCs, red blood cells; Hb, hemoglobin; MCH, mean corpuscular hemoglobin. Data are shown as means ± SEM of samples. The levels of iron variables at M0 were compared with those at M3 and M6. ** p* < 0.05. *** p* < 0.001.

**Figure 5 ijms-24-13779-f005:**
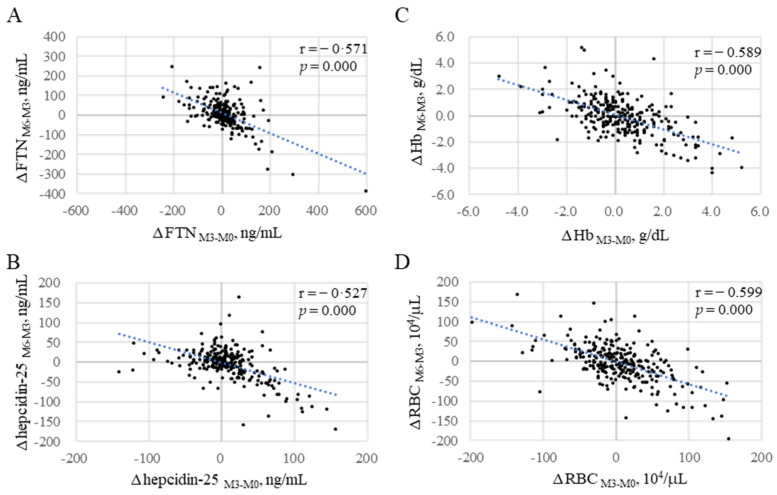
Inverse correlations between the changes in iron variables from M0 to M3 and those from M3 to M6 (black dots, *n* = 268). (**A**): ΔFTN_M3-M0_ vs. ΔFTN_M6-M3_; (**B**): ΔRBC_M3-M0_ vs. ΔRBC_M6-M3_; (**C**): ΔHb_M3-M0_ vs. ΔHb_M6-M3_; (**D**): ΔHEP_M3-M0_ vs. Δhepcidin-25_M6-M3_. Only 63 cases (23.5%) were positive for both ΔFTN_M3-M0_ and Δ FTN_M6-M3_. Δ_M3-M0_: Changes from M0 to M3; Δ_M6-M3_: changes from M3 to M6. FTN, ferritin; Hb, hemoglobin; RBCs, red blood cells. The blue dotted line describes the regression line.

**Figure 6 ijms-24-13779-f006:**
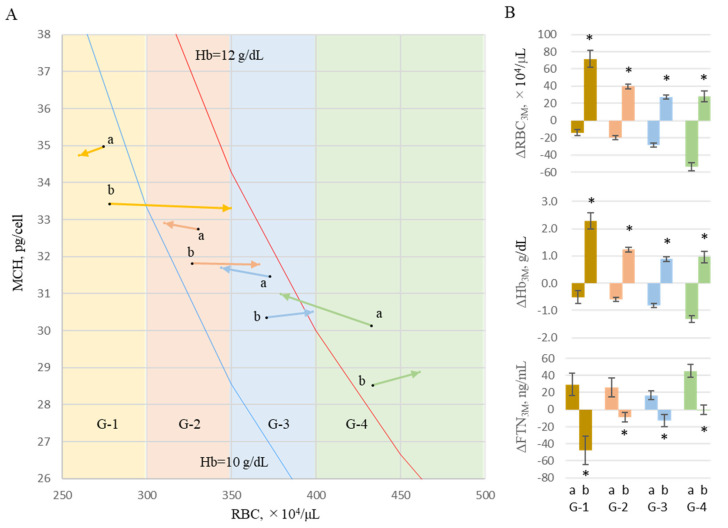
Effects of RBCs on changes in FTN. To analyze the effect of the change in RBC counts on FTN values over 3 months, the levels at M0 and M3 were treated as starting points for the first and second 3-month periods, and levels at M3 and M6 were treated as 3-month points, respectively. Two datasets of iron variables were evaluated equally, and the changes in iron variables over 3 months were represented as Δ_3M_ (*n* = 536). Each case was classified into 4 groups, G-1, G-2, G-3, and G-4, according to the RBC value at start points (M0 and M3): RBC ≤ 300, 300 < RBC ≤ 350, 350 < RBC ≤ 400 and RBC > 400 × 10^4^/μL, respectively. Furthermore, each case was classified into G-a when each ΔRBC_3M_ was negative and G-b when positive. (**A**) The start and end points were vectorized as the mean values of MCH and RBCs. Hb 10g/dL-line (blue) and Hb 12g/dL-line (red) are shown based on the formula Hb = RBC × MCH. (**B**) The levels of ΔFTN_3M_ were presented in each group. Data are shown as mean ± SEM of samples. RBCs, red blood cells; MCH, mean corpuscular hemoglobin; Hb, hemoglobin; FTN, ferritin. The levels of G-a were compared with those of G-b. ** p* < 0.05.

**Figure 7 ijms-24-13779-f007:**
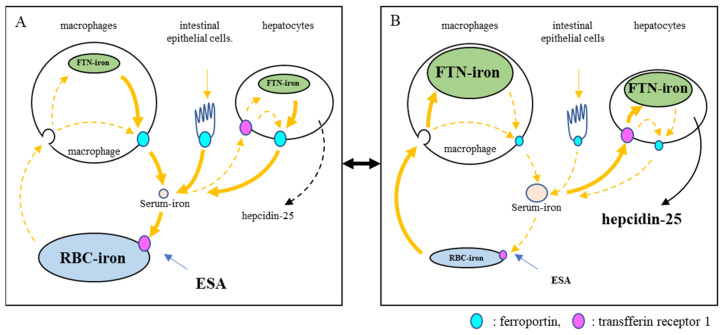
Model of crosstalk between the erythropoietic system and iron metabolic system. The stimulation of erythropoiesis by ESA is a trigger for crosstalk. When ESA stimulates hematopoiesis (**A**), serum iron is rapidly consumed and decreases. This triggers a decrease in the expression of hepcidin-25. Iron is then supplied from iron-store cells via FPN. If the supply of iron recovered from senescent red blood cells alone is inadequate, stored iron is consumed. This phenomenon appears to be a shift from FTN iron to RBC iron. In this situation, iron is readily absorbed from the intestinal tract. When erythropoiesis decreases due to the limitation of ESA (**B**), serum iron increases because of reduced iron consumption in bone marrow, and hepcidin-25 also increases. As a result, the iron supply to blood is suppressed, and unsupplied iron is stored in cells. Iron recovered from senescent red blood cells is also stored. This appears to be a shift from RBC iron to FTN iron. In this situation, iron absorption from the intestinal tract is suppressed. FTN, ferritin; ESA, erythropoiesis-stimulating agent; RBCs, red blood cells.

**Figure 8 ijms-24-13779-f008:**
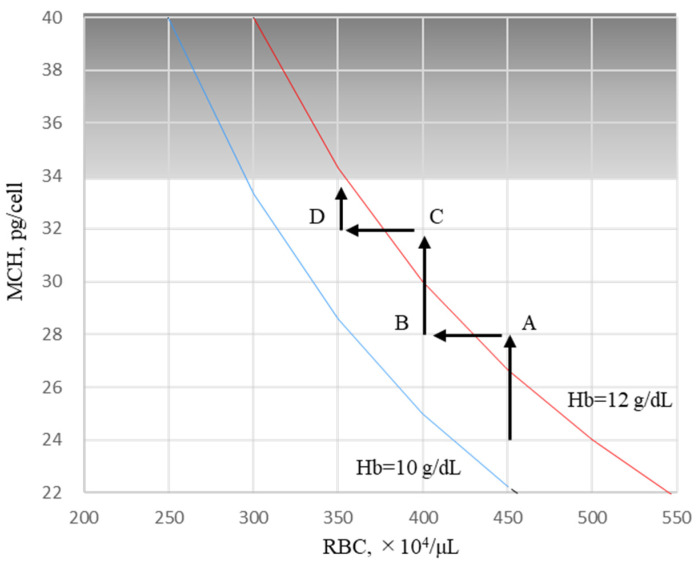
Hypothesis of Hb adjustment shown in the RBC/MCH diagram during FCH therapy. When the Hb level is ≥12 g/dL, ESA is arbitrarily reduced, and Hb levels are adjusted between 10 and 12 g/dL according to the guidelines. Hb level, which was 13 g/dL in RBC 450 × 10^4^/μL (MCH 28.9 pg, point A), was reduced to 11.6 g/dL when reducing ESA and resetting RBCs to 400 × 10^4^/μL (point B). During this period, the FTN level increased because RBC iron’s capacity decreased, and the corresponding amount of iron shifted to FTN iron. When FCH administration was continued, MCH increased with iron absorption, and Hb levels easily exceeded 12 g/dL (RBC to 400 × 10^4^/μL, MCH 32.5 pg, point C). If ESA was reduced again and RBCs reset to 350 × 10^4^/μL, Hb levels improved to 11.4 g/dL (point D) with another increase in the FTN level. When RBCs are ≤ 350 × 10^4^/μL, there is little risk of Hb being ≥12 g/dL even if FCH therapy is continued because there is an upper limit against MCH, which falls normally within the range of 27–33 pg. At this stage, there is no need to further reduce the ESA, and FTN reaches a plateau without increasing. Hb, hemoglobin; RBCs, red blood cells; MCH, mean corpuscular hemoglobin; FCH, ferric citrate hydrate; ESA, erythropoiesis-stimulating agent.

**Table 1 ijms-24-13779-t001:** Baseline characteristics of R-OIAT participants.

Characteristics	Baseline (*n* = 268)
Age (years)	63.0 (11.6)
Males, age, *n* (%)	63.8 (11.3), 153 (57%)
Females, age, *n* (%)	62.0 (12.0), 115 (43%)
Riona	
3 tablets (750 mg of FCH)	149 (55.6%)
6 tablets (1500 mg of FCH)	101 (37.7%)
9 tablets (2250 mg of FCH)	18 (6.7%)
ESA	
epoetin alfa/beta, *n* (%)	40 (14.9%)
darbepoetin alfa, *n* (%)	133 (49.6%)
epoetin beta pegol, *n* (%)	65 (24.3%)
No ESA, *n* (%)	30 (11.2%)

Data are mean (SD) or *n* (%). FCH, ferric citrate hydrate; ESA, erythropoiesis-stimulating agent.

**Table 2 ijms-24-13779-t002:** Characteristics of participants and iron status at the time of the iron absorption test.

	Baseline		3 Months		6 Months		
	M0 (*n* = 268)		M3 (*n* = 268)		M6 (*n* = 268)		
	Mean	(SD)	Mean	(SD)	Mean	(SD)	*p*
ESA, IU/week	3679.3	(3406.8)	3256.1	(3205.9)	3147.7	(3026.6)	0.000
RBCs, 10^4^/μL	360.3	(44.5)	364.9	(48.4)	360.9	(45.5)	0.301
Hb, g/dL	11.2	(1.2)	11.4	(1.3)	11.4	(1.2)	0.019
Ht, %	34.6	(3.8)	35.1	(4.0)	34.9	(3.6)	0.162
MCV	95.8	(8.8)	96.6	(5.5)	96.6	(7.5)	0.301
MCH, pg/cell	31.2	(2.2)	31.5	(2.1)	31.7	(2.0)	0.000
Plat, 10^4^/μL	19.2	(5.9)	20.1	(14.8)	18.8	(6.1)	0.018
S-Fe, μg/dL	65.7	(26.2)	70.3	(28.4)	69.8	(27.8)	0.013
FTN, ng/mL	100.7	(93.2)	108.9	(99.6)	116.7	(102.7)	0.000
TSAT, %	27.4	(11.6)	28.9	(12.0)	29.5	(12.5)	0.009
Hepcidin-25, ng/mL	42.9	(38.7)	50.5	(41.6)	45.6	(35.6)	0.000
ΔFe2h, μg/dL	26.6	(37.2)	24.7	(35.8)	22.9	(32.3)	0.062
P, mg/dL	5.5	(1.3)	5.5	(1.3)	5.5	(1.3)	0.982
Albumin, g/dL	3.9	(3.4)	4.1	(4.1)	3.9	(3.5)	0.127
AST, IU/L	13.5	(6.7)	13.8	(6.3)	13.9	(7.8)	0.663
ALT, IU/L	11.4	(5.4)	11.9	(7.6)	12.2	(9.4)	0.898
Al-P, IU/mL	239.5	(110.5)	241.7	(110.2)	238	(111.6)	0.134
γ-GTP, IU/L	21.3	(21.0)	20.7	(17.0)	22.2	(25.0)	0.455

The obtained data are expressed as the mean ± standard deviation (SD). A one-way repeated measures analysis of variance (ANOVA) was used for the comparison of data at M0, M3, and M6. The statistical significance was set at *p* < 0.05. ESA, erythropoiesis-stimulating agent; RBCs, red blood cells; Hb, hemoglobin; Ht, hematocrit; MCV, mean corpuscular volume; MCH, mean corpuscular hemoglobin; S-Fe, serum iron; FTN, ferritin; TSAT, transferrin saturation; ΔFe2h, iron absorption in 2 h; P, phosphate; AST, aspartate aminotransferase; ALT, alanine aminotransferase; Al-P, alkaline phosphatase; GTP, glutamyl transpeptidase.

**Table 3 ijms-24-13779-t003:** Effects of the amount of FCH on the changes in iron absorption and iron variables for 6 months.

		Time Course						F		
Iron Variable	FCH (mg)	M0		M3		M6		FCH	Time	Interaction
								Amounts	Course	Effects
ΔFe2h, μg/dL	750	23.0	(37.5)	18.5	(36.6)	17.2	(28.0)	7.58 *	2.42	0.35
	1500	29.9	(37.1)	29.6	(35.3)	29.1	(36.2)			
	2250	25.6	(44.9)	27.8	(50.2)	21.2	(46.5)			
FTN, ng/mL	750	72.8	(72.0)	81.3	(70.4)	84.5	(78.0)	22.02 **	7.92 *	0.168
	1500	136.7	(103.8)	147.9	(120.8)	160.3	(114.4)			
	2250	129.4	(112.3)	118.6	(105.2)	139.2	(121.8)			
Hepcidin-25, ng/mL	750	33.9	(32.2)	41.1	(32.8)	36.6	(24.5)	8.23 **	5.66 *	0.78
	1500	53.8	(43.1)	63.0	(49.6)	54.7	(41.9)			
	2250	56.4	(44.7)	58.9	(39.5)	68.8	(51.2)			
TSAT, %	750	26.5	(11.5)	29.1	(12.5)	28.7	(11.2)	0.52	5.13 *	0.60
	1500	28.4	(11.7)	28.4	(11.2)	29.9	(13.6)			
	2250	28.3	(12.1)	30.1	(12.2)	32.8	(15.4)			
RBCs, 10^4^/μL	750	360.3	(42.5)	361.2	(47.6)	358.2	(44.9)	5.29 *	0.38	0.71
	1500	365.3	(44.9)	372.5	(48.7)	368.7	(42.5)			
	2250	332.3	(50.3)	353.8	(50.9)	339.8	(58.8)			
Hb, g/dL	750	11.2	(1.1)	11.3	(1.2)	11.3	(1.2)	4.63 *	5.87 *	0.73
	1500	11.4	(1.2)	11.6	(1.4)	11.6	(1.1)			
	2250	10.6	(1.5)	11.3	(1.4)	11.1	(1.5)			
MCH, pg/cell	750	31.1	(2.3)	31.5	(2.2)	31.6	(2.2)	1.31	9.85 *	0.91
	1500	31.2	(2.0)	31.4	(1.8)	31.6	(1.8)			
	2250	32.1	(2.0)	32.2	(2.5)	32.3	(2.0)			
ESA, IU/week	750	3701	(3318)	3231	(3367)	3015	(3031)	1.36	0.77	3.20
	1500	3606	(3477)	3092	(2738)	3096	(2773)			
	2250	3902	(3901)	4375	(4134)	4527	(4052)			

The obtained data are expressed as the mean ± standard deviation (SD). A two-way (amount × time) repeated measures ANOVA was used for the comparison of the data at M0, M3, and M6. When a significant difference was detected, a multiple comparison test was performed using the post hoc Bonferroni correction. The statistical significance was set at *p* < 0.05. FCH = 750 mg (*n* = 149); FCH = 1500 mg (*n* = 101); FCH 2250 mg (*n* = 18). FCH, ferric citrate hydrate; ΔFe2h, iron absorption in 2 h; FTN, ferritin; TSAT, transferrin saturation; RBCs, red blood cells; Hb, hemoglobin; MCH, mean corpuscular hemoglobin; ESA, erythropoiesis-stimulating agent. * *p* < 0.05. ** *p* < 0.001.

**Table 4 ijms-24-13779-t004:** Predictors for ΔFe2h.

Parameter Estimates					
Variables	Log ΔFe2h, μg/dL		95% CI		*p* Value
	B	SE	Lower	Upper	
hepcidn-25, ng/mL	−0.459	0.094	−0.643	−0.276	0.000
RBCs, 10^4^/μL	0.989	0.719	−0.421	2.398	0.169
MCH, pg/cell	−1.220	1.582	−4.321	1.882	0.441
TSAT, %	−0.648	0.230	−1.099	−0.197	0.005
FTN, ng/mL	0.099	0.133	−0.161	0.359	0.457
ESA, IU/week	−0.188	0.121	−0.426	0.049	0.120
FCH, 750 mg	0.005	0.218	−0.431	0.422	0.983
FCH, 1500 mg	0.376	0.213	−0.041	0.794	0.077
FCH, 2250 mg	0 ^a^				
age	0.005	0.004	−0.002	0.013	0.155
sex, female	0.041	0.088	−0.131	0.214	0.637
sex, male	0 ^a^				

^a^ This parameter is redundant and is set to 0. ΔFe2h, iron absorption in 2 h; RBCs, red blood cells; MCH, mean corpuscular hemoglobin; TSAT, transferrin saturation; FTN, ferritin; ESA, erythropoiesis-stimulating agent; FCH, ferric citrate hydrate.

**Table 5 ijms-24-13779-t005:** Predictors for hepcidin-25 and FTN.

Parameter Estimates										
	Log Hepcidin-25, ng/mL		95% CI		*p* Value	Log Ferritin, ng/mL		95% CI		*p* Value
Variables	B	SE	Lower	Upper		B	SE	Lower	Upper	
hepcidin-25, ng/mL						0.297	0.030	0.239	0.355	0.000
FTN, ng/mL	0.754	0.060	0.637	0.871	0.000					
RBC, 10^4^/μL	0.790	0.312	0.178	1.401	0.011	−1.392	0.182	−1.749	−1.035	0.000
MCH, pg/cell	1.667	0.789	0.121	3.213	0.035	0.858	0.514	−0.150	1.866	0.095
TSAT, %	0.372	0.116	0.144	0.600	0.001	0.233	0.068	0.099	0.367	0.001
ESA, IU/week	−0.353	0.055	−0.462	−0.245	0.000	−0.084	0.045	−0.172	0.005	0.064
FCH, 750 mg	−0.141	0.056	−0.252	0.031	0.012	−0.101	0.068	−0.234	0.031	0.135
FCH, 1500 mg	−0.144	0.061	−0.263	−0.026	0.017	0.131	0.072	−0.009	0.270	0.067
FCH, 2250 mg	0 ^a^					0 ^a^				
age	0.000	0.001	−0.002	0.003	0.773	0.000	0.0015	−0.003	0.003	0.798
sex, female	0.130	0.036	0.060	0.200	0	−0.037	0.033	−0.101	0.028	0.263
sex, male	0 ^a^					0 ^a^				

^a^ This parameter is redundant and is set to 0. RBC, Red blood cell; MCH, mean corpuscular hemoglobin; TSAT, transferrin saturation; FTN, ferritin; ESA, erythropoiesis-stimulating agent; FCH, ferric citrate hydrate.

## Data Availability

Data is contained within the article. The data that support the findings of this study are available on request from the corresponding author, [N.T.]. The data are not publicly available due to their containing information that could compromise the privacy of research participants.

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
