# Peer review of "Oral Iron Absorption of Ferric Citrate Hydrate and Hepcidin-25 in Hemodialysis Patients: A Prospective, Multicenter, Observational Riona-Oral Iron Absorption Trial"

_ijms, 2023, doi:10.3390/ijms241813779_

Round 1
Reviewer 1 Report
This interesting article describes the variation of hepcidin , the master hormone of the iron metabolism in a prospective cohort of Japanese patients treated by ferric citrate as a phosphate bider together as oral iron therapy ; authors also analyzed classical iron biomarkers and capacity of iron absorption by the gut. This information is new, since hepcidine has not been previously measured nor followed in this situation to my knowledge.
I have two concerns :
1°Authors should discuss their results in light of the recent experimental findings in mice by Mark Hanudel and coworkers highly coherent with their study ((Enteral ferric citrate absorption is dependent on the iron transport protein ferroportin . Kidney International (2022) 101, 711–719 https://doi.org/10.1016/j.kint.2021.10.036)vb)
2° discussion should focus on their results and not to broadly discuss (without strict relationship)all the iron metabolism and all the dialysis.
this article has to be reviewed in deep and rewritten by an English or American native speaking translator
Author Response
Authors really appreciate your careful review.
- Authors should discuss their results in light of the recent experimental findings in mice by Mark Hanudel and coworkers highly coherent with their study ((Enteral ferric citrate absorption is dependent on the iron transport protein ferroportin . Kidney International (2022) 101, 711–719)
A: Thank you for introducing the literature. Authors cited the paper, which provides an important perspective for the discussion of our results.
Authors added Line 236-239: “More recently, Hanudel et al. [18] demonstrated that intestinal ferric citrate absorption relies on conventional enterocyte iron transport by ferroportin, without significant paracellular absorption, using Tmprss6 and FPN knockout mice.” Added sentences are highlighted in yellow in text.
- discussion should focus on their results and not to broadly discuss (without strict relationship) all the iron metabolism and all the dialysis.
A; Authors fully understand your point that discussion should be focused on the results.
Although I have expanded the scope of discussion because this Special Issue is "Recent Advance on Iron Metabolism, Ferritin and Hepcidin Research", authors are sure that readers will be interested in “ESA-induced iron storage capacity in RBC”, so authors added it as a hypothesis. In dyalisis field, ferritin-iron tends to be evaluated as a body stored iron, so we presented a hypothesis to emphasize that it is necessary to pay more attention to RBC and its behavior. It looks like crosstalk between iron metabolic system and erythropoietic syste (Figure 7, 8).
- Comments on the Quality of English Language
this article has to be reviewed in deep and rewritten by an English or American native speaking translator.
A; Our paper has undergone English language editing by MDPI. The text has been checked for correct use of grammar and commontechnical terms, and edited to a level suitable for rep orting research in a scholarly journal. MDPI uses experienced, native English speaking editors.
- 4. Authors corrected ml to mL, ml to mL and dl to dL in text, tables and figures. Corrections are highlighted in yellow in text.
- Authors corrected D to D in table 2 and 3. Corrections are highlighted in yellow in text.
- Authors corrected ferritin to abbreviation FTN in text, tables and figures. Corrections are highlighted in yellow in text.
- A reviewer pointed out an error in the analysis method, so the data were log-transformed before analysises.. No change in underlying trend. Table 2 to 5 were renewed. The authors added Line 421-422; “Iron variables were natural log-transformed to maintain a normal distribution. Added sentences are highlighted in yellow in text.

Reviewer 2 Report
It was a real pleasure reading your article that was focused on a subject of high interest for this area of expertise. As you have mentioned, oral ferrous iron administration is not recommended in hemodialysis patients due to several gastrointestinal side effects, and the use of FC increased in the last years, presenting less adverse effects in this group of patients. Therefore, your research could improve our current knowledge and the modality of using this therapy in daily practice. The methodology and the results of the study were clearly explained, and the conclusions were supported by your findings. The only suggestions are related to the abbreviations - please verify again the whole manuscript and explain them in the text when the abbreviations were firstly used (i.e. CRP, M0, M3, M6 etc.), and below the tables and figures, as well. In addition, please clarify if the included patients presented similar diet regarding daily iron and protein intake.
Author Response
We really appreciate your careful review.
- The only suggestions are related to the abbreviations - please verify again the whole manuscript and explain them in the text when the abbreviations were firstly used (i.e. CRP, M0, M3, M6 etc.), and below the tables and figures, as well.
A; As you pointed out, abbreviations were inappropriate. Authors explained all the abbreviations when they first appeared in text, tables and figures. Corrections are highlighted in yellow in text.
- please clarify if the included patients presented similar diet regarding daily iron and protein intake.
A; As you pointed out, the description about the meal was missing. The author added a description about it. Line 392-394: “The patient took a dialysis diet for 6 months of the trial, but did not take a special diet that determined the amount of protein and iron in the diet immediately before the iron absorption test.” Added sentences are highlighted in yellow in text.
- Authors corrected ml to mL, ml to mL and dl to dL in text, tables and figures. Corrections are highlighted in yellow in text.
- Authors corrected D to D in table 2 and 3. Corrections are highlighted in yellow in text.
- Authors corrected ferritin to abbreviation FTN in text, tables and figures. Corrections are highlighted in yellow in text.
- A reviewer pointed out an error in the analysis method, so the data were log-transformed before analysises.. No change in underlying trend. Table 2 to 5 were renewed. The authors added Line 421-422; “Iron variables were natural log-transformed to maintain a normal distribution. Added sentences are highlighted in yellow in text.

Reviewer 3 Report
In this multicenter study, the authors analyze iron metabolism during the therapy of phosphate-binding iron medication.
The main problem is the lack of assessment of compliance - perhaps the lack of ferritin increase was caused by lower ingested doses? In addition, the authors do not present AE related to the therapy. And changes in inflammatory status during the 6-month period. Therefore, it is difficult to analyze the fluctuation and responsible factors.
There is no information about blood transfusions. Any intravenous iron?
The presentation of the number of erythrocytes and platelets is strange and should be corrected - expressing erythrocytes in 10^6 and platelets in 10^3 / ml.
Some variables are not parametric - therefore before the analysis of correlation should be log-transformed (ferritin, hepcidin, delta 2h Fe).
Table 4 presents rather associates - and only 1 is significant - is it worth presenting this table?
If the therapy with phosphate-binding iron medication doses was unchanged during the 6-month study period it may be more reasonable to analyze only changes during the whole study period and divide the study group into 3 subgroups - ferritin increase > 20%; ferritin decrease > 20%, and unchanged +/-20%.
In my opinion, many of the associations are not important and make the paper unclear and difficult to follow. Certainly, the important are associations between hepcidin and ferritin, and 2h delta of iron with hepcidin.
Figure 5 does not bring any important message.
Table 5 is confusing and shows correlations without any physiological significance.
Please note that hemoglobin level is the most important variable (and not the number of erythrocytes).
Figure 6 has no sense. Effects of RBC on changes of FTN? As a consequence, it is difficult to explain.
The discussion is too much sophisticated. Please note that blood iron is related to hemoglobin level and not the number of erythrocytes.
In addition. it is well known that iron deficiency limits erythropoiesis. And that patients with low iron stores require larger ESA doses.
Therefore, the present conclusion is not acceptable.
Author Response
Authors really appreciate your careful review.
- The main problem is the lack of assessment of compliance - perhaps the lack of ferritin increase was caused by lower ingested doses? In addition, the authors do not present Advance Event related to the therapy. And changes in inflammatory status during the 6-month period. Therefore, it is difficult to analyze the fluctuation and responsible factors.
A: One tablet of FCH (250 mg) contains approximately 60 mg of ferric iron, so even 750mg of FCH is not lower ingested-doses.
As you indicated, Detail of AE related to the therapy were not presented in text. This study was an observational study without intervention, not a controlled study, so it is true that the description of AE is insufficient. Participants were a relatively stable group of patients who did not require changes in FCH during the 6-month follow-up period.
It is difficult to prove that there was no inflammation during the entire period, so patients were excluded if they had inflammation (CRP level > 0.5 mg/dL) on the day of the examination.
- There is no information about blood transfusions. Any intravenous iron?
A: Intravenous iron supplementation is included on exclusive list. 268 participants had not received blood transfusions for 6 months, as reported by their attending physicians. On Line 381, authors added description about blood transfusion.
- The presentation of the number of erythrocytes and platelets is strange and should be corrected - expressing erythrocytes in 10^6 and platelets in 10^3 / ml.
A: RBC 360×104/μL = 3.60×106/μL, Plat 19.2×104/μL =192×103/μL
- Some variables are not parametric - therefore before the analysis of correlation should be log-transformed (ferritin, hepcidin, delta 2h Fe).
A: Thank you for your advice. Authors confirmed that the data were not normally distributed, so they were log-transformed before analyses. Table 2 to 5 were renewed. The authors added Line 421-422; “Iron variables were natural log-transformed to maintain a normal distribution”. Added sentences are highlighted in yellow in text. Table 2 to 5 were renewed.
- Table 4 presents rather associates - and only 1 is significant - is it worth presenting this table?
A: DEE is essential for repeated data analysis of observational studies, and Table 4 showing the results; hepcidin-25 and TSAT is significant predictors, is also important. Authors appreciate the importance of table 4.
- If the therapy with phosphate-binding iron medication doses was unchanged during the 6-month study period it may be more reasonable to analyze only changes during the whole study period and divide the study group into 3 subgroups - ferritin increase > 20%; ferritin decrease > 20%, and unchanged +/-20%.
A: Your proposed method of classifying by the 6-month endpoint is also an important good method from a clinical point of view. This time, we started with the idea that if we could analyze why FTN rises and falls over a short period of 3 months, we might be able to understand the dynamics of iron. Therefore, authors analyzed changes in FTN over a short period of 3 months in Figs. 4 and 5 and arrived at the importance of red blood cells, which are large iron reservoirs.
- In my opinion, many of the associations are not important and make the paper unclear and difficult to follow. Certainly, the important are associations between hepcidin and ferritin, and 2h delta of iron with hepcidin.
A: Certainly, the important are associations between hepcidin and ferritin, and iron absorption with hepcidin. Here, the fact that the ESA was adjusted to keep Hb at 10-12 based on the guidelines was helpful in elucidating iron mobilization. In dialysis patients, ESA triggers hematopoietic oscillations and fluctuations in iron shift.
- Figure 5 does not bring any important message.
A: Authors think that the reverse fluctuations in the 3-month cycle are supporting evidence of following the guidelines. This is probably an artificial phenomenon that doctors are trying to keep at Hb 10-12 based on guidelines.
- Table 5 is confusing and shows correlations without any physiological significance.
A; I deleted the analysis on hepcidin-25 by GEE because it is hard to understand as you pointed out. However, authors believe that the prediction of FTN by GEE is important in considering the crosstalk of iron.
- Please note that hemoglobin level is the most important variable (and not the number of erythrocytes).
A: It goes without saying that Hb, which is involved in ATP production, is important, but when considering the movement of iron in iron metabolism, the size and number of RBC as a storehouse of iron that store Hb are also important. In other words, Hb is defined by RBC and RBC-Hb (MCH). We can separate the means of controlling these two factors. Hb is probably 12 g/dl for both RBC500 and MCH24, and RBC350 and MCH34, but the clinical perception is very different. Authors believe that understanding this situation is clinically important.
- Figure 6 has no sense. Effects of RBC on changes of FTN? As a consequence, it is difficult to explain.
A: The importance of RBC can be easily grasped as an image in the RBC/MCH diagram. No matter what region the RBCs are in, it shows that FTN decreases when RBC increases, and FTN increases when RBC decreases.
12.The discussion is too much sophisticated. Please note that blood iron is related to hemoglobin level and not the number of erythrocytes.
A; We need to be aware that when we use ESAs to stimulate hematopoiesis, we increase the capacity of iron stores, RBCs. This is an important viewpoint when considering crosstalk in iron metabolism. Please allow me to introduce this idea in this special issue. Authors believe that it will help to elucidate the mechanism of iron metabolism in the future.
- In addition. it is well known that iron deficiency limits erythropoiesis. And that patients with low iron stores require larger ESA doses.
A: It has been reported that myelosuppression with IDA is due to decreased EPOR as well as TfR2 in erythroblasts. The current proposal is not that iron deficiency inhibits hematopoiesis, but that when hematopoiesis is inhibited, iron shifts from RBC iron to FTN iron stores. Such a condition easily occurs in dialysis patients immediately after ESA administration, even without IDA.
- Therefore, the present conclusion is not acceptable.
A: RBCs and MCHs are rarely described in previous literature on dialysis patients treated with ESAs. This is because ESA is administered for the purpose of maintaining Hb. ESAs are drugs that stimulate hematopoiesis and increase red blood cell production. Why are RBCs not getting more attention? Of course, Hb is important. However, the era of guidelines emphasizing only Hb is coming to an end. It's from a little different point, but please let me blow a new wind.
